# The Role of Corporate Governance in Investment Efficiency and Financial Information Disclosure Risk in Companies Listed on the Tehran Stock Exchange

**Samira Moghaddamzadeh Kashani** 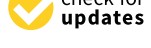 **and Mahmoud Mousavi Shiri \***

Department of Accounting, Payame Noor University (PNU), Tehran 9395-4697, Iran
* Correspondence: mousavi1973@pnu.ac.ir

**Abstract:** This study's primary purpose is to investigate corporate governance's role in investment efficiency and financial information disclosure risk in companies listed on the Tehran Stock Exchange. A multivariate linear regression model based on the panel data model was used to test the research hypotheses. The results of the survey of 140 companies listed on the Tehran Stock Exchange from 2015 to 2021 indicate that investment efficiency has increased by increasing the quality of corporate governance. In addition, research findings show that improving the quality of corporate governance reduces the risk of financial information disclosure. The life cycle and firm size were used to evaluate the robustness of the results obtained in this study. It was observed that improving corporate governance in companies in the stages of growth and maturity increases investment efficiency and reduces the financial information disclosure risk. In contrast, in companies that are in the decline stage, it reduces investment efficiency and increases the risk of financial information disclosure. In terms of firm size, it was also observed that, in small firms, as corporate governance increases, investment efficiency decreases, and the risk of financial information disclosure increases. However, investment efficiency and financial information disclosure reduce risk by improving large companies' corporate governance.

**Keywords:** investment efficiency; financial information disclosure risk; corporate governance

## 1. Introduction

Corporate governance has recently attracted much attention to protect investors' rights. It has become one of the essential mechanisms for managing and controlling companies in the capital market. At present, many countries have minimum corporate governance requirements. Although corporate governance indicators vary substantially among countries, different countries require listed companies on the stock exchange to comply with these requirements. Corporate governance minimizes the conflicting interests between internal and external stakeholders and shareholders. The corporate governance structure affects the quality of accounting disclosure and information quality assessment and guides analysts to accurately forecast future performance (Cheng et al. 2019). There is no consensus definition for corporate governance, but its ultimate goal is to achieve accountability, transparency, justice, fairness, and respect for the rights of all stakeholders (Firmansyah 2019). Corporate governance is not related to the primary operations of a company. Still, it is related to leading the company, monitoring the activities of the CEO, and assessing the accountability power of the company's executives to stakeholders. A proper corporate governance system can help companies gain investors' trust and encourage investment. According to the empirical research that has been conducted, implementing these principles at a company level will improve financial performance and increase company value (Huang et al. 2020).

Good corporate governance can provide a practical framework for balancing ownership and control, effective monitoring, appropriate incentives for the board and man-

agement to pursue goals for the company's interests and shareholders, and the equitable treatment of shareholders and other stakeholders (Peng et al. 2021). By establishing a quality corporate governance system, it is possible to take steps toward achieving a company's long-term goals by motivating its managers and employees (Black et al. 2010). Therefore, when corporate governance is appropriate, managers' behavior is expected to align with shareholders' interests (Med Bechir and Jouirou 2021). In general, corporate governance has a supervisory role over a company. Corporate governance mechanisms can maximize the interests of shareholders (predominantly minority shareholders) by monitoring the performance and behavior of managers and employees (Soliman 2020) and preventing the opportunistic behaviors of CEOs (Abd Karim et al. 2018).

Most of the research has confirmed the positive impact of corporate governance on performance (Braune et al. 2020; Paniagua et al. 2018; Iqbal et al. 2019; Mertzanis et al. 2019; Duppati et al. 2017). However, some studies have not reached such a conclusion (Rashed et al. 2018; Chen et al. 2017). Investment decisions in companies are determined by key factors such as macroeconomic factors, types of economic and monetary policies, money and capital markets, and corporate operations (Richardson 2006). In addition, managerial reasons such as unreasonable behavior of managers and inefficient financial markets in companies with a weak corporate governance system affect the companies' investment policies (Malmendier et al. 2011). It can be said that when a company has achieved a high level of investment efficiency, this ability indicates the strength of regulatory processes in the company, which has prevented inefficient investments of excess cash flow in the company. Through preventive monitoring of corporate governance over high-risk investment events, a company can detect an inefficient investment in the organization before it occurs, which can prevent an inefficient investment (He et al. 2019) because investment efficiency generally means investing in projects with a positive net present value, regardless of investing in projects with a negative net present value (Verdi 2006). Gompers et al. (2010) showed that the company's value would be less in a weak corporate governance system. This is why the ownership structure plays a significant role in the value of the company, because the lower the concentration of ownership, the more managers sacrifice some of the company's values to protect their interests, and as a result, the efficiency of the investment decreases. Managers have a strong incentive to invest in negative net present value (NPV) projects in companies with high free cash flows, especially when management monitoring is weak, where agency costs are high and corporate ownership and control are separated (Stulz 1990). Therefore, due to the infancy of corporate governance law in the Tehran Stock Exchange, evaluation of the impact of corporate governance on operational efficiency in the Tehran Stock Exchange market can demonstrate its effectiveness in this market.

In general, disclosure is the provision of the minimum legal requirement of information. Thus far, several structures, such as appropriateness, comprehensiveness, reliability, informativeness and timeliness, have been used as proxies of disclosure quality. The stronger the corporate governance system, the higher the information disclosure and the greater the information transparency (Braune et al. 2020). When corporate governance is weak, managers can increase financial reporting risk by manipulating financial information to achieve personal goals and, consequently, create information inefficiencies in determining stock prices in the capital market. According to the expectation of implementing corporate governance, corporate governance can affect the desired and more transparent disclosure of information and not allow managers to hide their poor performance with poor operational efficiency through undesirable disclosure of information. Investors consider corporate financial information as one of the sources to reduce information asymmetry. To evaluate the efficiency of the investment, investors refer to the financial information of companies. The more reliable and relevant the information is, the more possible it is to make the right decisions. The corporate governance system can prevent opportunistic behaviors of management in not disclosing sufficient and quality information by increasing management monitoring processes. Companies with better corporate governance have high-quality financial information disclosure, and providing high-quality financial

information convinces investors and other stakeholders to invest (Kouki and Attia 2016). Establishing a strong corporate governance mechanism can encourage investors to purchase the company's stock at a higher price by increasing the proper disclosure of financial information and gaining investors' trust, eventually maximizing the company's value. In Europe, having investigated the reasons for disclosing intangible capital information, it was concluded that improving corporate governance, which is subject to voluntary disclosure of information, causes an increase in corporate financial performance (Braune et al. 2020).

The high risk of financial information disclosure can coincide with a company's investment performance. Due to opportunistic behaviors of management, increasing the risk of financial information disclosure can indicate the inefficiency of the company's investments. An increase in the level of high-quality disclosure related to financial statements leads to a higher level of investment in a company (Li et al. 2019). It can also indicate a two-way relationship between the level of disclosure and the efficiency of investment in companies (Elberry and Hussainey 2020). Roychowdhury et al. (2019), in their study, have shown that the quality of reporting has a direct and positive effect on investment efficiency. In addition, information transparency is positively related to investment efficiency for firms with strong governance. By improving corporate governance, the quality of information disclosure and investment efficiency increases (Chen et al. 2021). The high quality of financial information disclosure in a company, which is influenced by the strong corporate governance structure in the company through more supervision and strict rules to control the actions of managers, and thereof the increasing of investment efficiency, is achieved through limiting an overinvestment in the negative NPV projects or under-investment by neglecting positive NPV ones (Elberry and Hussainey 2021). Good corporate governance can reduce information asymmetry, agency costs, and information search costs and can increase information transparency since corporate governance allows investors to experience fewer investment errors; sound corporate governance can ensure that while a company's managers have the incentive to make their own profits, they attempt to increase the interests of investors and the firm value (Cheng et al. 2019).

Corporate governance and its importance are relatively new issues in Iran. In 2004, corporate governance, based on OECD guidelines, gained public attention with the first attempt by the Tehran Stock Exchange to codify the first draft of corporate governance bylaws. In 2008, the OECD's corporate governance principles were translated into Persian. In 2010, the Securities and Exchange Commission (SEO) completed and formally approved the corporate governance regulations, but its implementation in companies has not yet been mandatory. During this time, several seminars, conferences, and awareness-raising activities on corporate governance were held. Meanwhile, the SEO attempted to improve corporate governance through separate regulations such as disclosure and transparency.

Due to the low quality of disclosure in the Iranian capital market compared to developed countries, the desirability of domestic and foreign investors to make new investments in the Iranian capital market has been challenged (Mehrabani 2012). In addition, unlike the majority of shareholders, the interests of minority shareholders are not protected, unlike in other countries where non-controlling shareholders sometimes have significant influence. No Iranian institution ranks companies based on characteristics such as returns, revenue, total assets, number of employees, etc., and Iran's internal control supervision mechanisms are inadequate. As a result, managers often prefer personal and corporate interests (Irani and Safari Gerayeli 2017). The existence of suitable conditions for profit-seeking managers will cause problems of agency theory in companies, in which corporate governance is expected to play a significant role in reducing agency costs.

Although the corporate governance literature is fully developed and many studies have been conducted in this field, no studies have been conducted on the impact of corporate governance on investment efficiency (as one of the essential roles of company leadership) and the impact of corporate governance on the risk of financial disclosure (as one of the criteria for determining the expected return of shareholders) that can indicate the effectiveness of the presence of corporate governance in the management of the company,

which can be one of the points that investors consider. Therefore, this study tries to answer whether the implementation of corporate governance legislation in Iran has affected the efficiency and risk of financial information disclosure of companies. Empirical evidence obtained in this regard can provide feedback on implementing corporate governance legislation for the Tehran Stock Exchange and investors in this market.

## 2. Literature Review

With the recent financial scandals around the world, many researchers have paid more attention to studying the effectiveness of corporate governance. Although much research has been conducted in developed economies, few studies have been conducted in emerging economies where corporate governance is newly evolving. Several types of research have been conducted in the field of corporate governance in developing countries, which have reached several results by surveying some mechanisms of corporate governance, such as the audit committee, size, the number of meetings, and financial expertise. The research conducted in Islamic banks showed that although the audit committee can limit the risk-taking of banks and supervise them for high-risk investments, it does not have such an effect on Islamic banks, and the effect of the Shariah committee is stronger on limiting these banks. Therefore, increasing the improvement of corporate governance mechanisms such as Shariah committees in Islamic banks will limit risk-taking in these types of banks (Nguyen 2021). By focusing on the audit committee structure's effectiveness, the bank's stability is enhanced through increased performance and investment levels (Nguyen 2022). Islamic banks with strong corporate governance (including smaller boards, independent boards, financial expertise and frequency of Shariah committee meetings) can reduce the Shariah non-compliant risk (Basiruddin and Ahmed 2019). The weaker the corporate governance mechanisms, including ownership structure, board structure, stakeholder's rights, and company relations, the more credit risk increases and financial stability decreases (Ballester et al. 2020). Companies with higher investment efficiency seem to have a stronger regulatory system because the high efficiency indicates a strong regulatory system that has increased the efficiency of investments. In addition, strong corporate governance seems to lead to higher quality financial information disclosure and, therefore, less risk of financial information disclosure.

### 2.1. Corporate Governance and Investment Efficiency

Investment efficiency means the organization invests in projects with positive net present value (Verdi 2006). The basic principles of corporate governance include ensuring the observance of ethics and protection of the stakeholders' rights, ensuring the observance of the code of ethics and other values. According to stakeholder theory, corporate governance should take into account the interests of all organizational stakeholders, increase moral obligations in an organization, and raise the responsibility of the individual to stakeholders by disclosing the information risk to stakeholders to help them make decisions, maintain wealth, and increase trust between the company and stakeholders (Habbash 2017). Ethical decision making and ethical values are fully reflected in issues such as conflicts of interest, opportunities to participate in fair transactions, gaining trust, correct use of a company's assets, operating per rules and laws, and encouragement in dealing with unethical practices (Li et al. 2019). A corporate governance system can help companies gain investors' trust; when corporate governance is appropriate, managers' behavior is expected to be in line with the interests of the shareholder. In other words, corporate governance leads to an increase in the value of a company (Black et al. 2006). Research conducted in China showed that managers in companies that have weak internal control over financial reporting are more likely to invest inefficiently (Lai et al. 2020).

The efficiency of investing in companies that have a stronger monitoring system is higher. According to agency theory, managers seek to maximize their interests in a company. In their own interests, they prefer to invest free cash flow in projects that may even be unprofitable. When the quality of corporate governance is strengthened, investment



efficiency improves. High investment efficiency can indicate a strong monitoring system in a company; this system causes investment efficiency to increase. Through corporate governance's preventive monitoring of high-risk investment events, they can identify an ineffective investment in the organization before it occurs; this identification can prevent an inadequate investment (He et al. 2019). Several research studies on corporate governance and investment efficiency concluded that there is a positive and significant relationship between corporate governance and financial performance (Mertzanis et al. 2019; Med Bechir and Jouirou 2021; Li et al. 2020). Some of these research studies have shown that the board's independence positively affects the relationship between corporate governance and financial performance (Paniagua et al. 2018; Al-ahdal et al. 2020). In other words, a strong corporate governance system improves financial performance (Duppati et al. 2017; Iqbal et al. 2019; Machmud et al. 2020; Srivastava and Kathuria 2020; Peng et al. 2021; Sheikh and Alom 2021), and the efficiency of resource investment has a positive and significant effect on the growth of financial performance (Özbuğday et al. 2020).

Soliman (2020) concluded in his study that, by having high audit quality and reducing information asymmetry, corporate governance has a positive effect on increasing the attraction of new investments and leads to an increase in the volume of investment in companies, and as a result, the value of the company increases. On the other hand, Rodrigues et al. (2020) concluded that investment efficiency has a positive and significant relationship with corporate governance mechanisms that help align the interests of managers and shareholders. Chen et al. (2016) showed that companies with positive free cash flow overinvest, while some corporate governance characteristics, such as larger board size, reduce overinvestment.

The studies conducted in the field of corporate governance and excessive managerial entrenchment and firm performance show that excessive managerial entrenchment reduces board monitoring and deteriorates the firm valuation in the capital markets because as CEOs become entrenched, they gain more control and seek to maximize their benefits rather than the interests of the shareholders. As a result, excessive managerial entrenchment has a negative impact on the shareholders' welfare. It causes a decrease in the efficiency of the company's performance and, subsequently, a reduction in the firm value (Antounian et al. 2021). On the other hand, the conclusions of the studies conducted in the field of corporate governance quality, leverage, and performance indicate that the higher the quality of corporate governance, the lower leverage (Memon et al. 2019), and as a result, financial performance increases (Zhou et al. 2021).

Wang et al. (2021) showed that investment efficiency increases with stock concentration through increasing corporate governance, such as reducing agency conflict, and according to the research conducted by Zhang (2020), earning informativeness decreases with the reduction in controlling of shareholder's ownership, and reducing agency conflict has a positive effect on improving the investment efficiency of companies. In addition, considering the recent crisis and the COVID-19 pandemic, studies (Hsu and Liao 2022) showed that good corporate governance could positively affect the stock market's performance in this era. Some corporate governance mechanisms, such as ownership structure, are strongly related to stock price reactions during the COVID-19 pandemic. Large companies and governments experience less stock price declines in response to the pandemic (Ding et al. 2021).

**Hypothesis 1.** *There is a positive and significant relationship between corporate governance and investment efficiency.*

### 2.2. Corporate Governance and Financial Information Disclosure Risk

The term disclosure, in its broadest concept, means providing information. Accountants use this term in a more limited way, meaning publishing financial information related to a company in financial reports (usually in annual reports). In the narrowest concept, information disclosure includes management discussions and analyses, footnotes to financial statements and supplementary financial statements (Clark 2016). Today, information

plays a significant role in the investment decisions of investors. Therefore, to obtain more information and to solve the problem of information asymmetry, information disclosure is used due to the separation of companies' ownership and management (Habbash et al. 2016).

With the need to disclose financial information for investors, studies have investigated the characteristics and deficiencies related to the disclosure of risks in companies' annual reports. By increasing the quality of disclosure achieved through monitoring and the composition and structure of the board, information asymmetry is reduced between a company and investors (Ghouma et al. 2018). Empirical evidence related to the mechanism of corporate governance and the quality of disclosure indicates that corporate governance plays a vital role in the quality of disclosure (Alagla 2019) such that some corporate governance mechanisms, such as board size, audit type, the audit committee independence, have a positive and significant effect on the financial reporting quality (Paul et al. 2018).

Even the effect of corporate governance on the voluntary disclosure of financial information has been proven in various research studies, indicating the company's desire to reduce information asymmetry and the risk of financial information disclosure. Al-Nimer (2019), while investigating the effect of corporate governance on voluntary disclosure, concluded that corporate governance mechanisms, including the board size and the audit committee size, have a positive and significant relationship with the level of voluntary disclosure. Shan (2019) reached a similar conclusion in another study and concluded that corporate governance mechanisms such as foreign ownership, the ratio of independent directors, and the age of a company promote voluntary disclosure. Lokman et al. (2012), by studying the relationship between the corporate governance quality and the voluntary disclosure of financial information, concluded that there is a higher probability of voluntary disclosure of information in companies with a high quality of corporate governance.

El-Deeb and Elsharkawy (2019) investigated the effect of board characteristics as one of the corporate governance mechanisms and disclosure. This article measures corporate governance mechanisms such as board independence, board size, CEO duality, and board gender diversity. The research results indicate that the auditor type and the board size have a significant and positive relationship with information disclosure. Jacoby et al. (2019) showed that both direct and indirect effects of internal corporate governance mechanisms, such as incentive compensation and board independence, increase the company's information transparency. Both corporate governance mechanisms and external control mechanisms help boost such transparency.

Li et al. (2020) showed that the risk of financial information disclosure has a positive and significant relationship with investment efficiency. This means that investment efficiency is improved by increasing financial information transparency; in addition, increasing the level of transparency when companies invest effectively acts as a positive signal for shareholders, and as a result, a high level of financial reporting disclosure is associated with an increase in investment efficiency (Elberry and Hussainey 2020; Biddle et al. 2009). In addition, other research shows that reporting quality directly and positively affects investment efficiency and provides better identification (Roychowdhury et al. 2019).

According to the theoretical foundations and the background of the study, and since financial information disclosure can be the reason for the higher efficiency of investment in companies, in the second hypothesis, the relationship between corporate governance and its effect on the risk of financial information disclosure is investigated.

**Hypothesis 2.** *There is a negative and significant relationship between corporate governance and financial information disclosure risk.*

### 3. Research Methodology

This study is considered a semi-experimental study in the field of accounting proof research, and in terms of its purpose, it is applied research. An inductive method was applied to the ex-post data (using historical data), and correlation analysis was used for the statistical analysis. The companies' real and audited information has been analyzed and

consolidated using a multivariate linear regression model. The required data for the study were extracted from Iranian databases, including RahvardNavin software and financial statements on the Codal website, and were matched with the information of financial statements. Excel spreadsheet software was used to prepare the information, and EVIEWS version 9 software was used for statistical analysis.

### 3.1. Population and Statistical Sample

The study's statistical population includes 140 Tehran Stock Exchange companies and 980 company-year observations from 2015 to 2021. According to the research period (2015–2021), the companies must be available on the Tehran Stock Exchange (TSE) before 2015, and their names should not have been removed from the listed companies by the end of 2021. Financial companies, banks, and investment companies are not included in the selected companies in this study. In addition, the required information of these companies, such as corporate governance information, is available to the researcher. Their shares have been bought and sold continuously on the stock exchange market without a trading break (the trading break was less than three months). Since the fourteen principles to improve corporate governance in companies admitted to the Tehran Stock Exchange were implemented in 2015, the period of this study was considered from 2015 to 2021.

### 3.2. Research Models for Hypothesis Testing

In this study, corporate governance was surveyed from the perspective of investment efficiency and the financial information disclosure risk at the company level, and the independent and dependent variables were measured as follows:

In this study, corporate governance ($CG_{it}$) is an independent variable, and firstly, indicators for corporate governance should be defined to measure the quality of corporate governance. In this study, 10 indicators were used to identify corporate governance quality, as described in Table 1. All of these indicators have been measured as artificial variables (zero and one variables). (Li et al. 2020)

**Table 1.** Indicators for identifying the level of corporate governance quality.

| Variable | Description | Reference |
|---|---|---|
| Separation of the position of the CEO from the member of the board (CEO-duality) ($DUAL_{it}$) | If the CEO holds the chairman or vice chairman of the board, the number is one; otherwise, the number is zero (there is no duality). | Polovina and Peasnell (2020) |
| Non-executive board of director ($BODS_{it}$) | For firms with a more non-executive board of directors than the executive board of directors, we put the number one, and for other firms, we put the number zero. | Kieschnick and Moussawi (2018) |
| Institutional ownership percentage ($INSTOWN_{it}$) | For firms with institutional ownership by ownership percentage greater than 20%, the number is one; for other firms, the number is zero. | Vijayakumaran and Vijayakumaran (2019) |
| The concentration of ownership ($CONCENOWN_{it}$) | If there is at least one real shareholder in a firm with more than 10% shares, the number is one; otherwise, it is zero. | Gillani Ahmad et al. (2018) |
| Main or subsidiary of a firm ($PARENT_{it}$) | If a firm is part of the group firms, the number should be one; otherwise, the number should be zero. | Polovina and Peasnell (2020) |
| Existence of an internal audit function ($IA_{it}$) | If a firm has an internal audit function, the number is one; otherwise, it is zero. | Al-Baidhani (2014) |
| Free-floating shares' percentage ($FLOATSHARE_{it}$) | If a company's free-floating shares' percentage is greater than 20%, we will consider the number one, and for the rest of the firms, it is zero. | Khan (2019) |
| Firm audit ($AUDIT_{it}$) | If the firm's auditor is an auditing organization (the largest auditing organization in Iran), the number is one; otherwise, it is zero. | Soliman (2020) |
| Type of audit opinion ($OPINION_{it}$) | If the type of audit opinion is an unadjusted report, the number is one; otherwise, it is zero | Awadallah (2020) |
| Existence of an audit committee ($AuditCommittee_{it}$) | If the non-executive members of the audit committee are more than the executive members, the number is one; otherwise, it is zero. | Arslan and Alqatan (2020) |

To measure the quality of corporate governance for each company year, we obtain the sum of the above 10 proxies and divide it by 10:

$$CG_{it} = \frac{\sum_{j=1}^{m} d_j}{\sum_{j=1}^{n} H_j}$$

where $\sum_{j=1}^{m} d_j$ is the sum of the virtual indicators obtained for each year of the company, and $\sum_{j=1}^{n} H_j$ is equal to the number of indicators considered for measuring corporate governance. The higher the value of the corporate governance ability measure, the higher (stronger) the quality of corporate governance, and vice versa; the lower this value of the measure is, the lower (weaker) the quality of corporate governance.

The variable of investment efficiency ($INVEFF_{it}$) is the dependent variable, which will be used in this study following Richardson's (2006) research:

Model 1:

$$I_{it} = \beta_0 + \beta_1 TOBINQ_{it-1} + \beta_2 CASH_{it-1} + \beta_3 AGE_{it-1} + \beta_4 SIZE_{it-1} + \beta_5 LEV_{it-1}$$
$$+\beta_6 RETURN_{it-1} + \beta_7 I_{it-1} + \varepsilon_{it}$$

where:

1.  $I_{it}$ = The ratio of a company's investment (investment in fixed and intangible assets and long-term investments) to the book value of the company's assets in year t.
2.  $TOBINQ_{it-1}$ = Tobin's Q ratio in year t − 1, which is equal to the ratio of the market value of assets (the book value of liabilities plus the market value of equity) divided by the book value of assets.
3.  $CASH_{it-1}$ = Cash holding ratio in year t − 1, which is equal to the ratio of cash flow plus short-term investments divided by the book value of assets.
4.  $AGE_{it-1}$ = The firm age in year t − 1, which is equal to the natural logarithm of the firm's life since its establishment.
5.  $SIZE_{it-1}$ = The firm size in year t − 1, which is equal to the natural logarithm of the book value of the firm's assets.
6.  $LEV_{it-1}$ = The financial leverage of the company in year t − 1, which is equal to the ratio of the book value of liabilities to assets.
7.  $RETURN_{it-1}$ = The company's annual stock return rate in year t − 1, which is equal to the annual stock return rate extracted from the Codal website in Iran.
8.  $I_{it-1}$ = The amount of the company's investment in year t − 1 (investment in fixed and intangible assets and long-term investments) to the book value of the company's assets in year t − 1.
9.  $\varepsilon_{it}$ = Positive residuals (positive deviation from the expected investment) indicate the selection of projects with a negative net present value or overinvestment ($OverINV_{it}$), and negative residuals (negative deviation from the expected investment) indicate the passing of investment opportunities with a positive net present value or underinvestment ($UnderINV_{it}$). The absolute value of the regression equation's residuals is an inverse index of investment efficiency, i.e., investment inefficiency. The lower this variable is, the lower the inefficiency (higher efficiency), and the higher this number is, the higher the inefficiency (lower efficiency). Therefore, to solve this problem, we multiply the absolute value of the error by a negative number.

The financial information disclosure risk ($FIDR_{it}$) is a dependent variable based on the absolute value of the residual error's standard deviation, estimated using the modified Jones model. Based on Kothari et al. (2005), we measure the cross-sectional discretionary accruals by using the modified Jones model and define the relationship between the total accruals with variables such as sales, gross property, plant and equipment, and return on assets for a specific period:

$$\left(\frac{TA_{it}}{A_{i(t-1)}}\right) = \alpha_1 \left(\frac{1}{A_{i(t-1)}}\right) + \alpha_2 \left(\frac{\Delta SAL_{it}}{A_{i(t-1)}}\right) + \alpha_1 \left(\frac{PPE_{it}}{A_{i(t-1)}}\right) + \alpha_4 ROA_{i,t} + \varepsilon_{it}$$

where:

1. $A_{i(t-1)}$ = The total assets of the firm in year t − 1;
2. $\Delta SAL_{it}$ = The difference in sales between year t − 1 and year t;
3. $PPE_{it}$ = Gross property, plant and equipment;
4. $ROA_{it}$ = Return on assets;
5. $\varepsilon_{it}$ = Estimation error;
6. $\alpha_1.\alpha_2. \alpha_3.\alpha_4$ = Firm's specific parameters.

In this equation, $TA_{it}$ is defined as the total accruals, which can be calculated as follows:

$$TA_{it} = (\Delta CA_{it} - \Delta CASH_{it}) - (\Delta CL_{it} + \Delta STD_{it}) - DAE_{it}$$

where:

1. $\Delta CA_{it}$ = The firm's change in current assets between year t − 1 and year t;
2. $\Delta CASH_{it}$ = The firm's change in cash flow between year t − 1 and year t;
3. $\Delta CL_{it}$ = The firm's change in current liabilities between year t − 1 and year t;
4. $\Delta STD_{it}$ = The firm's change in the short-term debt of long-term liabilities between year t − 1 and year t;
5. $DAE_{it}$ = Firm's depreciation and amortization expense of tangible and intangible assets in year t.

The non-discretionary accrual $NDA_{it}$ are calculated as follows for the estimation period:

$$NDA_{it} = \alpha_1 \left(\frac{1}{A_{i(t-1)}}\right) + \alpha_2 \left(\frac{\Delta SAL_{it} - \Delta AR_{it}}{A_{i(t-1)}}\right) + \alpha_3 \left(\frac{PPE_{it}}{A_{i(t-1)}}\right) + \alpha_4 ROA_{it}$$

In the above equation, $\Delta AR_{it}$ is defined as the firm's change in net accounts receivable between year t − 1 and year t.

Finally, discretionary accruals are obtained according to the following equation:

$$DA_{it} = \frac{TA_{it}}{A_{i(t-1)}} - NDA_{it}$$

In order to calculate the financial information disclosure risk, we use the standard deviation of the absolute residual error from three years ago.

The method of measuring the control variables is described in Table 2:

**Table 2.** Control variables of the research.

| Variable | Description |
|---|---|
| Rate of Return on Assets ($ROA_{it}$) | It is the net profit after tax ratio to the book value of the firm i's assets in year t. |
| Inventory ($INVENT_{it}$) | It is the ratio of inventory to the book value of the firm i's assets in year t. |
| notes and Accounts Receivable ($RECEV_{it}$) | It is the ratio of notes and accounts receivable to the book value of the firm i's assets in year t. |
| Firm Age ($AGE_{it}$) | It is the natural logarithm of the firm's age from the date of establishment to the year under survey. |
| Cash Flow from Operations ($CFO_{it}$) | It is the ratio of cash flow from operation to the book value of the firm i's assets in year t. |
| Property, Plant and Equipment ($PPE_{it}$) | It is the property, plant and equipment ratio to the book value of the company i's assets in year t. |
| Tobin's Q Ratio ($TOBINQ_{it}$) | It is the ratio of the market value of assets (the market value of assets is equal to the book value of debt plus the market value of equity) to the book value of the assets of firm i in year t. |
| Leverage ($LEV_{it}$) | It is the ratio of debt to the book value of assets of firm i in year t. |
| Firm Size ($SIZE_{it}$) | It is the natural logarithm of net sales of firm i in year t. |

In this study, the following two models were used to investigate the relationship between corporate governance and investment efficiency and the financial information disclosure risk:

Model 1: Test of the first hypothesis.

$$INVEFF_{it} = \beta_0 + \beta_1 CG_{it} + \beta_2 ROA_{it} + \beta_3 INVENT_{it} + \beta_4 RECEV_{it} + \beta_5 AGE_{it}$$
$$+ \beta_6 CFO_{it} + \beta_7 PPE_{it} + \beta_8 TOBINQ_{it} + \beta_9 LEV_{it} + \beta_{10} SIZE_{it}$$
$$+ \varepsilon_{it}$$

Model 2: Test of the second hypothesis.

$$FIDR_{it} = \beta_0 + \beta_1 CG_{it} + \beta_2 ROA_{it} + \beta_3 INVENT_{it} + \beta_4 RECEV_{it} + \beta_5 AGE_{it}$$
$$+ \beta_6 CFO_{it} + \beta_7 PPE_{it} + \beta_8 TOBINQ_{it} + \beta_9 LEV_{it} + \beta_{10} SIZE_{it}$$
$$+ \varepsilon_{it}$$

## 4. Data Analysis

### 4.1. Descriptive Statistics

The descriptive statistics results of variables are presented in Tables 3 and 4, which show the descriptive parameters for each variable separately. These parameters mainly include information related to central tendencies, such as minimum, maximum, mean, and median, as well as information related to dispersion index, such as standard deviation. The most crucialcentral tendency is the mean, which indicates the balance point of the distribution (the center of gravity) and is a suitable index to show the centrality of the data.

**Table 3.** Descriptive statistics of the research variables.

| Variable | Symbol | Mean | Median | SD | Min | Max |
|---|---|---|---|---|---|---|
| Corporate Governance | $CG_{it}$ | 0.545 | 0.500 | 0.122 | 0.200 | 0.900 |
| Investment Efficiency | $INVEFF_{it}$ | −0.061 | −0.029 | −0.094 | −0.000 | −0.528 |
| Financial Information Disclosure Risk | $FIDR_{it}$ | 0.189 | 0.166 | 0.148 | 0.011 | 0.890 |
| Rate of Return on Assets | $ROA_{it}$ | 0.063 | 0.059 | 0.147 | −0.466 | 0.434 |
| The ratio of Inventory to Assets | $INVENT_{it}$ | 0.238 | 0.220 | 0.126 | 0.0007 | 0.695 |
| The ratio of notes and Accounts Receivable to Assets | $RECEV_{it}$ | 0.305 | 0.272 | 0.181 | 0.002 | 0.854 |
| Natural Logarithm of Firm Age | $AGE_{it}$ | 3.656 | 3.761 | 0.347 | 2.708 | 4.219 |
| The ratio of Cash Flow from Operations to Assets | $CFO_{it}$ | 0.102 | 0.092 | 0.119 | −0.228 | 0.475 |
| The ratio of Property, Plant and Equipment to Assets | $PPE_{it}$ | 0.253 | 0.210 | 0.177 | 0.003 | 0.849 |
| Tobin's Q Ratio | $TOBINQ_{it}$ | 1.738 | 1.486 | 1.166 | 0.568 | 28.202 |
| Leverage | $LEV_{it}$ | 0.659 | 0.661 | 0.202 | 0.188 | 0.987 |
| Firm Size | $SIZE_{it}$ | 13.809 | 13.800 | 1.475 | 2.564 | 19.566 |

**Table 4.** Descriptive statistics of the investment efficiency variables.

| Variable | Symbol | Mean | Median | SD | Min | Max | Observations |
|---|---|---|---|---|---|---|---|
| Previous Year's Tobin's Q | $Q_{it-1}$ | 1.694 | 1.445 | 1.132 | −0.568 | 28.202 | 980 |
| The ratio of the Previous year's Cash Holding Value | $CASH_{it-1}$ | 0.051 | 0.031 | 0.061 | 0.000 | 0.479 | 980 |
| Previous Year's Firm Age | $AGE_{it-1}$ | 3.629 | 3.737 | 0.356 | 2.639 | 4.639 | 980 |
| Previous Year's Firm Size | $SIZE_{it-1}$ | 14.074 | 13.910 | 1.349 | 10.504 | 19.149 | 980 |
| Previous Year's Leverage | $LEV_{it-1}$ | 0.656 | 0.664 | 0.191 | 0.198 | 0.955 | 980 |
| Rate of Previous Year's Stock Return | $ROA_{it-1}$ | 0.470 | 0.165 | 0.922 | −0.562 | 4.548 | 980 |
| The ratio of the Previous Year's Investment | $Invest_{it-1}$ | 0.036 | 0.010 | 0.105 | −0.241 | 0.493 | 980 |

According to the results of Table 3, the mean value of the corporate governance variable is approximately 0.545, and the median value is 0.500, with a minimum value of 0.200 and a maximum value of 0.900, which has a standard deviation of about 0.122. This shows that the average corporate governance score in the surveyed companies was approximately 54.5%, which signifies the success of the surveyed companies in obtaining

half of the corporate governance score on average. The mean of the investment efficiency variable is approximately −0.061, and the median is −0.029, with a minimum value of −0.0003 and a maximum value of −0.528, with a standard deviation of about 0.094. The tiny distance between the mean and the standard deviation of the investment efficiency variable indicates the normality of this variable. The mean value of the financial information disclosure risk variable is approximately 0.189, and the median is 0.166, with a minimum value of 0.011 and a maximum value of 0.890, with a standard deviation of about 0.148. The great distance between the mean and the standard deviation implies the considerable difference in the financial information disclosure in the surveyed companies.

In Table 4, the mean of the previous year's Tobin's Q is approximately 1.694. The median is 1.445, with a minimum value of −0.568 and a maximum value of 28.202, with a standard deviation of about 1.132, indicating that, on average, the market value of assets was about 1.6% of their book value in the surveyed companies. Moreover, the mean (standard deviation) of the previous year's investment ratio variable is approximately 0.036 (0.105), implying that, on average, the amount of investments was equal to 3.6% of the book value of assets. In addition, by comparing the investment ratio of the current year and the previous year, the mean of the investment ratio in year t is about 0.034, while the ratio of investment to assets in year t − 1 is about 0.036, indicating that the investment level of companies has decreased. The mean of the previous year's cash holding ratio is approximately 0.051, the median is 0.031, with a minimum value of 0.0004 and a maximum value of 0.479, which has a standard deviation of about 0.061, and this signifies that the amount of cash flow plus the previous year's short-term investment was about 5.1% of the previous year's book value of the assets.

### 4.2. Inferential Statistics

Multivariate linear regression is used for hypotheses testing. The data can be time series, cross-sectional, or combined with choosing a data analysis model. Considering that the data of this study are of a panel data type, it should be determined whether it is a panel data type or a combination type. Therefore, the Chow test was used. For observations whose test probability is more than 5%, or in other words, their test statistic is less than the table statistic, the combined model is used, and for observations whose test probability is less than 5%, the panel data model will be used to estimate the model. The panel data model can be conducted using two "random effect" and "fixed effect" models. To select the most appropriate model, the Hausman test was used. The observations whose test probability is less than 5% are used from the fixed effect model, and those whose test probability is more than 5% are used from the random effect model to estimate the model.

The results of these tests are presented in Tables 5 and 6, and according to the obtained significance level and to the acceptable significance level, which is less than 5%, the results indicate that the panel data model should be used for the study models considering the fixed effects.

**Table 5.** Chow test results.

| Model | Test Type | F Statistic | $p$ Value Significance | Test Results |
|-------|-----------|-------------|------------------------|--------------|
| Efficiency | Selecting the panel data model | 1.687 | 0.000 | The panel data model is preferable |
| (1) | Selecting the panel data model | 9.170 | 0.000 | The panel data model is preferable |
| (2) | Selecting the panel data model | 9.133 | 0.000 | The panel data model is preferable |

**Table 6.** Hausman test results.

| Model | Test Type | X2 Statistic | *p* Value Significance | Test Results |
|---|---|---|---|---|
| Efficiency | Selecting the fixed effect or random effect model | 203.952 | 0.000 | Fixed effect model is preferable |
| (1) | Selecting the fixed effect or random effect model | 17.669 | 0.061 | The random effect model is preferable |
| (2) | Selecting the fixed effect or random effect model | 16.573 | 0.084 | The random effect model is preferable |

The investment efficiency model estimation results are presented in Table 7.

**Table 7.** Results of the statistical test of the investment efficiency model.

| $Invest_{it} = \gamma_0 + \gamma_1 Q_{it-1} + \gamma_2 CASH_{it-1} + \gamma_3 AGE_{it-1} + \gamma_4 SIZE_{it-1} + \gamma_5 LEV_{it-1} + \gamma_6 RETURN_{it-1} + \gamma_7 Invest_{it-1} + \varepsilon_{it}$ | | | | | |
|---|---|---|---|---|---|
| **Dependent Variable: The Firm's Investment Ratio** | | | | | |
| Variable | Coefficient | Std. Error | T-Statistic | Sig. | VIF |
| $C$ | −0.090 | 0.188 | −4.217 | 0.000 | - |
| $Q_{it-1}$ | 0.010 | 0.190 | 5.731 | 0.000 | 1.064 |
| $CASH_{it-1}$ | 0.073 | 0.142 | 3.043 | 0.002 | 1.073 |
| $AGE_{it-1}$ | 0.009 | 0.201 | 2.703 | 0.007 | 1.040 |
| $SIZE_{it-1}$ | 0.004 | 0.009 | 3.451 | 0.000 | 1.070 |
| $LEV_{it-1}$ | −0.010 | 0.211 | −1.768 | 0.077 | 1.089 |
| $RETURN_{it-1}$ | 0.000 | 0.001 | 0.476 | 0.634 | 1.014 |
| $Invest_{it-1}$ | 0.085 | 0.027 | 3.064 | 0.002 | 1.051 |
| Adjusted R2 | 0.085 | | F-statistic | | 14.097 |
| Durbin–Watson statistic | 1.995 | | Significance | | 0.000 |

As shown in Table 7, according to the obtained F statistic (14.097) and its significance level (0.000), it can be claimed that the investment efficiency model has high significance at the 95% confidence level. In addition, the variable standard deviation of the ratio of the previous year's investment of companies is 0.027, and the previous year's cash holding value is 0.142.

*4.3. The Results of the Hypothesis Test*

**Hypothesis 1.** *There is a positive and significant relationship between corporate governance and investment efficiency.*

As shown in Table 8, the corporate governance variable coefficient (CGit) coefficient is 0.083. The T-statistic is equal to 3.184, which is significant at 0.001. Since it is less than the prediction error (5%), the significance of the independent variable is confirmed at a confidence level of more than 95%. This result shows a positive and significant relationship between corporate governance and investment efficiency in both periods. In other words, investment efficiency is also higher in companies with a higher corporate governance quality.

**Table 8.** Results of the statistical test of the first hypothesis.

| INVEFF = β0 + β1 CG + β2 ROA+ β3 INVENT+ β4 RECEV+ β5 AGE+ β6 CFO+ β7 PPE+ β8 TOBINQ+ β9 LEV+ β10 SIZE + ε | | | | | |
|---|---|---|---|---|---|
| **Dependent Variable: Investment Efficiency (INVEFF)** | | | | | |
| **Independent Variable: Corporate Governance (CG)** | | | | | |
| **Variable** | **Coefficient** | **Std. Error** | **T-Statistic** | **Sig.** | **VIF** |
| $C$ | 0.121 | 0.043 | 2.807 | 0.005 | − |
| $CG_{it}$ | 0.083 | 0.026 | 3.184 | 0.001 | 1.187 |
| $ROA_{it}$ | 0.082 | 0.025 | 3.246 | 0.001 | 2.296 |
| $INVENT_{it}$ | 0.000 | 0.024 | 0.006 | 0.995 | 1.450 |
| $RECEV_{it}$ | −0.090 | 0.017 | −5.035 | 0.000 | 1.980 |
| $AGE_{it}$ | 0.065 | 0.007 | 9.081 | 0.000 | 1.039 |
| $CFO_{it}$ | −0.011 | 0.024 | −0.474 | 0.368 | 1.346 |
| $PPE_{it}$ | −0.076 | 0.018 | −4.066 | 0.000 | 1.918 |
| $TOBINQ_{it}$ | 0.013 | 0.003 | 4.098 | 0.000 | 1.153 |
| $LEV_{it}$ | −0.074 | 0.017 | −4.181 | 0.000 | 1.966 |
| $SIZE_{it}$ | 0.018 | 0.001 | 9.571 | 0.000 | 1.215 |
| Adjusted R2 | 0.275 | | F-statistic | | 38.268 |
| Durbin–Watson statistic | 1.724 | | Significance | | 0.000 |

As the results show in Table 8, there is a positive and significant relationship between the variables of return on assets ($ROA_{it}$), firm age ($AGE_{it}$), Tobin's Q ratio ($TOBINQ_{it}$), and firm size ($SIZE_{it}$) with investment efficiency ($INVEFF_{it}$). In addition, the results of the study showed that there is a negative and significant relationship between the variables of notes receivable ($RECEV_{it}$), property, plant and equipment ($PPE_{it}$), and leverage ($LEV_{it}$) with investment efficiency ($INVEFF_{it}$). There is no significant relationship between the inventory variable ($INVENT_{it}$) and cash flow from operations ($CFO_{it}$) with investment efficiency ($INVEFF_{it}$). Moreover, the F-statistic (Fisher's statistic) amount is 38.268, and its significance level is 0.000; since the significance level is below 5% error, the multivariate linear regression model is significant, and the adjusted R-squared coefficient is 0.275. The adjusted R-squared coefficient in multivariate linear regression is a coefficient that indicates the amount of changes in the dependent variable by the independent and control variables. Its value is between zero and one, and in Table 8, about 27.5% of the investment efficiency changes are explained by the independent variable (corporate governance) and control variables.

**Hypothesis 2.** *There is a negative and significant relationship between corporate governance and financial information disclosure risk.*

As shown in Table 9, the corporate governance variable coefficient ($CG_{it}$) is −0.085. The T-statistic is equal to −4.513, which is significant at 0.000. Since it is less than the prediction error (5%), the significance of the independent variable is confirmed at a confidence level of more than 95%. This result shows a negative and significant relationship between corporate governance and financial information disclosure risk. In other words, the risk of financial information disclosure is higher in companies where the quality of corporate governance is lower.

**Table 9.** Results of the statistical test of the second hypothesis.

| FIDR = β0 + β1 CG + β2 ROA+ β3 INVENT+ β4 RECEV+β5 AGE+ β6 CFO+ β7 PPE+ β8 TOBINQ+ β9 LEV+ β10 SIZE + ε | | | | | |
|---|---|---|---|---|---|
| Dependent Variable: Financial Information Disclosure Risk (FIDR) | | | | | |
| Independent Variable: Corporate Governance (CG) | | | | | |
| Variable | Coefficient | Std. Error | T-Statistic | Sig. | VIF |
| $C$ | 0.119 | 0.044 | 2.689 | 0.007 | - |
| $CG_{it}$ | −0.085 | 0.018 | −4.513 | 0.000 | 1.022 |
| $ROA_{it}$ | 0.088 | 0.025 | 3.522 | 0.000 | 2.294 |
| $INVENT_{it}$ | −0.001 | 0.025 | −0.055 | 0.955 | 1.433 |
| $RECEV_{it}$ | −0.091 | 0.018 | −4.879 | 0.000 | 1.980 |
| $AGE_{it}$ | 0.066 | 0.007 | 9.044 | 0.000 | 1.040 |
| $CFO_{it}$ | 0.002 | 0.024 | 0.109 | 0.912 | 1.327 |
| $PPE_{it}$ | 0.058 | 0.016 | 3.455 | 0.000 | 1.833 |
| $TOBINQ_{it}$ | 0.012 | 0.003 | 3.756 | 0.000 | 1.113 |
| $LEV_{it}$ | −0.068 | 0.017 | −3.895 | 0.000 | 1.977 |
| $SIZE_{it}$ | 0.017 | 0.002 | 8.784 | 0.000 | 1.221 |
| Adjusted R2 | 0.269 | | F-statistic | 37.091 | |
| Durbin–Watson statistic | 1.721 | | Significance | 0.000 | |

As the results show in Table 9, there is a positive and significant relationship between the variables of return on assets ($ROA_{it}$), firm age ($AGE_{it}$), Tobin's Q ratio ($TOBINQ_{it}$), property, plant and equipment ($PPE_{it}$), and firm size ($SIZE_{it}$) with the financial information disclosure risk ($FIDR_{it}$). In addition, the results of the study showed that there is a negative and significant relationship between the variables of notes receivable ($RECEV_{it}$) and leverage ($LEV_{it}$) with the financial information disclosure risk ($FIDR_{it}$). There is no significant relationship between the inventory variable ($INVENT_{it}$) and cash flow from operations ($CFO_{it}$) with the financial information disclosure risk ($FIDR_{it}$). Moreover, the amount of F-statistic (Fisher's statistic) is 37.091, and its significance level is 0.000; since the significance level is below 5% error, the multivariate linear regression model is significant, and the adjusted R-squared coefficient amount is 0.269. The adjusted R-squared coefficient in multivariate linear regression is a coefficient thatindicates the amount of changes in the dependent variable by the independent and control variables, and its value is between zero and one; in Table 9, about 26.9% of the financial information disclosure risk's changes are explained by the independent variable (corporate governance) and control variables.

*4.4. Robustness Test*

To evaluate the robustness of our results, we also surveyed our hypotheses by considering factors such as the life cycle and the firm size.

4.4.1. The Role of the Life Cycle

According to the studies, companies cannot maintain investment efficiency when they move through different life cycle stages. Ahmed et al. (2020) showed that the companies' investment efficiency is lower during the decline stage and has the highest level of investment efficiency during the growth and maturity stages. Therefore, to evaluate the robustness of the hypotheses results, we split our samples into three categories based on their life cycle (growing, maturing, and declining companies) according to Dickinson's (2011) method.

As the results show in Table 10, the corporate governance variables' coefficients in companies in the two stages of growth and maturity are 0.383 and 2.377, respectively. Their significance is less than the error level of 0.05. Therefore, the positive coefficients of these variables imply that corporate governance improves the investment efficiency in companies in both growth and maturity stages. In contrast, the relationship between

corporate governance and investment efficiency is negative and significant in companies in the decline stage.

**Table 10.** The results of the statistical test of the first hypothesis in the life cycle stages.

| INVEFF = β0 + β1 CG + β2 ROA+ β3 INVENT+ β4 RECEV+β5 AGE+ β6 CFO+ β7 PPE+ β8 TOBINQ+ β9 LEV+ β10 SIZE + ε | | | | | | | | | |
|---|---|---|---|---|---|---|---|---|---|
| Dependent Variable: Investment Efficiency (INVEFF) and Independent Variable: Corporate Governance (CG) | | | | | | | | | |
| | Growth Stage | | | Maturity Stage | | | Decline Stage | | |
| Variable | Coefficient | T-Statistic | Sig. | Coefficient | T-Statistic | Sig. | Coefficient | T-Statistic | Sig. |
| $C$ | 0.174 | 5.463 | 0.000 | 0.377 | 6.844 | 0.000 | 0.375 | 3.112 | 0.001 |
| $CG_{it}$ | 2.377 | 5.016 | 0.000 | 0.383 | 9.142 | 0.000 | −0.144 | −5.182 | 0.000 |
| $ROA_{it}$ | 0.015 | 3.158 | 0.001 | 0.072 | 1.908 | 0.057 | 1.204 | 2.128 | 0.033 |
| $INVENT_{it}$ | −0.043 | −0.415 | 0.677 | −0.001 | −0.698 | 0.484 | −0.003 | −0.851 | 0.394 |
| $RECEV_{it}$ | −0.091 | −2.287 | 0.002 | −0.393 | −3.497 | 0.001 | −1.492 | −2.988 | 0.004 |
| $AGE_{it}$ | 0.072 | 1.908 | 0.057 | 0.011 | 0.142 | 0.011 | 0.361 | 3.008 | 0.002 |
| $CFO_{it}$ | −0.100 | −0.759 | 0.447 | −0.029 | −0.242 | 0.808 | −0.016 | −0.787 | 0.432 |
| $PPE_{it}$ | −0.003 | −3.590 | 0.000 | −0.023 | −1.703 | 0.088 | −0.022 | −2.897 | 0.003 |
| $TOBINQ_{it}$ | 0.069 | 2.336 | 0.020 | 0.060 | 3.151 | 0.001 | 0.133 | 2.926 | 0.004 |
| $LEV_{it}$ | −0.016 | −2.215 | 0.027 | −0.084 | −4.640 | 0.004 | −0.095 | −2.678 | 0.008 |
| $SIZE_{it}$ | 0.185 | 1.788 | 0.074 | 0.157 | 5.096 | 0.000 | 0.096 | 2.876 | 0.004 |
| Adjusted R2 | 0.358 | | | 0.322 | | | 0.316 | | |
| Durbin–Watson statistic | 1.845 | | | 1.897 | | | 1.967 | | |
| F-statistic | 9.158 | | | 6.197 | | | 4.923 | | |
| Significance | 0.000 | | | 0.000 | | | 0.000 | | |

The Wald test was used to check the results obtained in this step. The significance of the difference in the adjusted R-squared coefficient of the regression models in the three life cycle stages was tested through the Wald test, and the results are presented in Table 11.

**Table 11.** Survey the significance of the difference in the adjusted R-squared coefficient of the regression models according to the firms' life cycle for the first hypothesis (Wald test).

| Comparison of Explanatory Power | Wald F-Statistic | Sig. | Test Result |
|---|---|---|---|
| Growth stage and maturity stage | 0.602 | 0.438 | There is no significant difference between the adjusted R-squared coefficient of growth and maturity stages |
| Growth stage and decline stage | 3.172 | 0.042 | There is a significant difference between the adjusted R-squared coefficient of growth and decline stages |
| Maturity stage and decline stage | 6.153 | 0.013 | There is a significant difference between the adjusted R-squared coefficient of maturity and decline stages |

According to the results reported in Table 12, to test the robustness of the second hypothesis, the coefficients of corporate governance variable in companies that are in two stages of growth and maturity are −0.017 and −1.167, respectively, and their significance is less than the error level of 0.05. Therefore, the positive coefficients of these variables indicate a negative and significant relationship between corporate governance and financial information disclosure risk in companies in both growth and maturity stages. In contrast, in companies in the decline stage, there is a positive and significant relationship between corporate governance and financial information disclosure risk; in companies with higher corporate governance, the risk of financial information disclosure is higher.

**Table 12.** The results of the statistical test of the second hypothesis in the life cycle stages.

| FIDR = β0 + β1 CG + β2 ROA+ β3 INVENT+ β4 RECEV+ β5 AGE+ β6 CFO+ β7 PPE+ β8 TOBINQ+ β9 LEV+ β10 SIZE + ε | | | | | | | | | |
|---|---|---|---|---|---|---|---|---|---|
| Dependent Variable: Financial Information Disclosure Risk (FIDR) and Independent Variable: Corporate Governance (CG) | | | | | | | | | |
| | Growth Stage | | | Maturity Stage | | | Decline Stage | | |
| Variable | Coefficient | T-Statistic | Sig. | Coefficient | T-Statistic | Sig. | Coefficient | T-Statistic | Sig. |
| $C$ | 0.272 | 2.925 | 0.004 | 2.373 | 4.358 | 0.000 | 0.172 | 3.345 | 0.000 |
| $CG_{it}$ | −0.017 | −4.995 | 0.000 | −1.167 | −5.770 | 0.000 | 0.432 | 4.294 | 0.000 |
| $ROA_{it}$ | 0.017 | 1.804 | 0.071 | 0.194 | 3.603 | 0.000 | 0.008 | 2.297 | 0.023 |
| $INVENT_{it}$ | −0.933 | −0.676 | 0.499 | −0.022 | −0.669 | 0.503 | −0.026 | −1.445 | 0.149 |
| $RECEV_{it}$ | −0.007 | −2.840 | 0.004 | −0.085 | −3.281 | 0.003 | −0.175 | −0.267 | 0.788 |
| $AGE_{it}$ | 1.204 | 2.128 | 0.033 | 2.853 | 3.676 | 0.001 | 0.008 | 2.684 | 0.007 |
| $CFO_{it}$ | −0.124 | −0.736 | 0.461 | −1.851 | −0.566 | 0.571 | −0.005 | −1.945 | 0.052 |
| $PPE_{it}$ | 0.095 | 2.678 | 0.008 | 1.057 | 2.920 | 0.004 | 0.332 | 3.281 | 0.001 |
| $TOBINQ_{it}$ | 0.058 | 3.184 | 0.001 | 0.112 | 2.809 | 0.005 | 0.215 | 3.135 | 0.001 |
| $LEV_{it}$ | −0.031 | −1.868 | 0.062 | −0.048 | −3.086 | 0.000 | −0.010 | −2.473 | 0.014 |
| $SIZE_{it}$ | 0.088 | 2.108 | 0.026 | 0.806 | 8.025 | 0.001 | 0.034 | 1.901 | 0.057 |
| Adjusted R2 | 0.497 | | | 0.448 | | | 0.458 | | |
| Durbin–Watson statistic | 1.358 | | | 1.578 | | | 1.807 | | |
| F-statistic | 7.285 | | | 4.578 | | | 2.485 | | |
| Significance | 0.000 | | | 0.000 | | | 0.000 | | |

The Wald test results to survey the life cycle's role in the relationship between corporate governance and financial information disclosure risk are presented in Table 13.

**Table 13.** Survey the significance of the difference in the adjusted R-squared coefficient of the regressions according to the firms' life cycle in the second hypothesis (Wald test).

| Comparison of Explanatory Power | Wald F-Statistic | Sig. | Test Result |
|---|---|---|---|
| Growth stage and maturity stage | 1.052 | 0.292 | There is no significant difference between the adjusted R-squared coefficient of growth and maturity stages |
| Growth stage and decline stage | 3.257 | 0.039 | There is a significant difference between the adjusted R-squared coefficient of growth and decline stages |
| Maturity stage and decline stage | 6.045 | 0.014 | There is a significant difference between the adjusted R-squared coefficient of maturity and decline stages |

4.4.2. The Role of Firm Size

Madhani (2016) found that large firms have better corporate governance practices and information disclosure than small firms, as large firms provide higher information disclosure than their smaller peer firms. Research results show that firm size affects the performance of the companies (Luo et al. 2022). For the robustness of the results for the first and second hypotheses, we divided the surveyed firms into two categories, large and small, based on their assets. Based on corporate governance, we measured the investment efficiency and the financial information disclosure risk.

According to the results obtained in this study, which can be seen in Table 14, the coefficient of corporate governance in small firms is −0.401. It indicates a negative and significant relationship between investment efficiency and corporate governance. In contrast, the coefficient of corporate governance in large firms is positive and 0.437, which implies a positive and significant relationship between the two mentioned variables.

**Table 14.** The results of the statistical test for the first hypothesis in small and large firms.

| INVEFF = β0 + β1 CG + β2 ROA+ β3 INVENT+ β4 RECEV+ β5 AGE+ β6 CFO+ β7 PPE+ β8 TOBINQ+ β9 LEV+ β10 SIZE + ε | | | | | | |
|---|---|---|---|---|---|---|
| **Dependent variable: Investment Efficiency (INVEFF) and Independent variable: Corporate Governance (CG)** | | | | | | |
| | **Small Firms** | | | **Large Firms** | | |
| **Variable** | **Coefficient** | **T-Statistic** | **Sig.** | **Coefficient** | **T-Statistic** | **Sig.** |
| $C$ | 0.037 | 3.782 | 0.004 | 0.644 | 83.453 | 0.000 |
| $CG_{it}$ | −0.401 | −2.432 | 0.015 | 0.437 | 14.711 | 0.000 |
| $ROA_{it}$ | 0.698 | 2.468 | 0.016 | 1.145 | 2.040 | 0.041 |
| $INVENT_{it}$ | −0.004 | −0.698 | 0.485 | −0.055 | −0.693 | 0.488 |
| $RECEV_{it}$ | −0.118 | −4.033 | 0.000 | −1.943 | −2.136 | 0.042 |
| $AGE_{it}$ | 0.205 | 3.316 | 0.001 | 2.755 | 2.605 | 0.015 |
| $CFO_{it}$ | −0.048 | −0.196 | 0.847 | −0.046 | −0.295 | 0.768 |
| $PPE_{it}$ | −4.247 | −2.308 | 0.021 | −0.209 | −1.680 | 0.094 |
| $TOBINQ_{it}$ | 0.040 | 3.520 | 0.001 | 0.113 | 1.889 | 0.058 |
| $LEV_{it}$ | −0.091 | −2.022 | 0.044 | −0.072 | −2.232 | 0.026 |
| $SIZE_{it}$ | 0.008 | 2.590 | 0.099 | 0.069 | 2.336 | 0.020 |
| Adjusted R2 | 0.181 | | | 0.225 | | |
| Durbin–Watson statistic | 2.850 | | | 2.162 | | |
| F-statistic | 8.614 | | | 11.027 | | |
| Significance | 0.000 | | | 0.000 | | |

The results reported in Table 15 indicate that the second hypothesis is a negative and significant relationship between large firms and a positive and significant relationship between small firms. This means that the higher the corporate governance in large firms, the lower the financial information disclosure risk. In small firms, the higher the corporate governance, the higher the risk of financial information disclosure.

**Table 15.** The results of the statistical test for the second hypothesis in small and large firms.

| FIDR = β0 + β1 CG + β2 ROA+ β3 INVENT+ β4 RECEV+ β5 AGE+ β6 CFO+ β7 PPE+ β8 TOBINQ+ β9 LEV+ β10 SIZE + ε | | | | | | |
|---|---|---|---|---|---|---|
| **Dependent Variable: Financial Information Disclosure Risk (FIDR) and Independent Variable: Corporate Governance (CG)** | | | | | | |
| | **Small Firms** | | | **Large Firms** | | |
| **Variable** | **Coefficient** | **T-Statistic** | **Sig.** | **Coefficient** | **T-Statistic** | **Sig.** |
| $C$ | 0.018 | 2.954 | 0.003 | 0.877 | 3.712 | 0.001 |
| $CG_{it}$ | 0.086 | 3.576 | 0.000 | −0.757 | −25.265 | 0.000 |
| $ROA_{it}$ | 0.086 | 5.965 | 0.000 | 0.096 | 2.876 | 0.004 |
| $INVENT_{it}$ | −0.301 | −0.348 | 0.531 | −0.005 | −3.995 | 0.000 |
| $RECEV_{it}$ | −0.009 | −3.097 | 0.002 | −0.272 | −2.670 | 0.007 |
| $AGE_{it}$ | 1.418 | 3.244 | 0.001 | 1.266 | 4.717 | 0.000 |
| $CFO_{it}$ | −0.055 | −1.627 | 0.147 | −0.038 | −1.075 | 0.283 |
| $PPE_{it}$ | −0.132 | −0.933 | 0.350 | −0.033 | −0.825 | 0.410 |
| $TOBINQ_{it}$ | 4.506 | 1.779 | 0.075 | 0.010 | 3.785 | 0.000 |
| $LEV_{it}$ | −0.163 | −2.388 | 0.017 | −0.042 | −3.019 | 0.002 |
| $SIZE_{it}$ | 0.174 | 3.345 | 0.000 | 0.064 | 8.915 | 0.000 |
| Adjusted R2 | 0.267 | | | 0.152 | | |
| Durbin–Watson statistic | 1.845 | | | 1.909 | | |
| F-statistic | 13.475 | | | 7.135 | | |
| Significance | 0.000 | | | 0.000 | | |

The significance of the difference in the adjusted R-squared coefficient of the regression models according to the firm size was tested through the Wald test, and the results are presented in Table 16.

**Table 16.** Survey of the significance of the difference in the adjusted R-squared coefficient of the regressions according to the firm size in the first and second hypothesis (Wald test).

| Comparison of Explanatory Power | Wald F-Statistic | Sig. | Test Result |
|---|---|---|---|
| Small and large firms in the model (1) | 3.358 | 0.048 | There is a significant difference between the adjusted R-squared coefficients |
| Small and large firms in the model (2) | 2.683 | 0.007 | There is a significant difference between the adjusted R-squared coefficients |

## 5. Discussion and Conclusions

In recent years, corporate governance has attracted extensive academic attention in many majors, most of which have found that certain aspects of a company's corporate governance are related to its corporate performance or financial position. In addition, currently, hardly anyone realizes the importance of investment efficiency in companies and the quality of financial information disclosure because investors and all creditors make their financial decisions based on financial information. They always need transparent information about an organization's financial performance to make decisions. Investment efficiency and financial information disclosure quality will have the highest efficiency in a system with strong corporate governance. Corporate governance also affects an organization's financial performance and can be called a protector for minority shareholders. Since weak corporate governance is one of the negative aspects of an organization, which causes the disclosure of information in a vague and non-transparent manner, corporate governance is vital and attractive to investors. Kouki and Attia (2016) showed that financial information disclosure would be less in a weak corporate governance system. When corporate governance is weak, management's ability to be transparent and disclose financial information is weakened. By increasing the investment efficiency and the quality of financial information disclosure, it is possible to establish a high-quality corporate governance system to achieve long-term goals by motivating the managers and employees of a company (Black et al. 2010).

This study investigates the relationship between corporate governance, investment efficiency, and information financial disclosure risk. The first hypothesis showed a positive and significant relationship between corporate governance and investment efficiency. In other words, investment efficiency is also higher in companies with a higher quality of corporate governance. This result implies that companies with higher investment efficiency have a stronger monitoring system. According to agency theory, managers seek to maximize their interests in a company. They prefer to spend free cash flow on investment projects, which sometimes may even be a loss. Managers invest free cash resources in projects with a negative NPV. When the investment efficiency increases, it seems that the company has had a high quality of corporate governance because the high investment efficiency indicates a strong monitoring system in a company, which has increased the investment efficiency. In addition, during the COVID-19 crisis, some corporate governance mechanisms, such as the size of the board, improved the company's performance (Khatib and Nour 2021). Therefore, the first hypothesis is confirmed in a positive direction.

The results of the second hypothesis indicated a negative and significant relationship between corporate governance and financial information disclosure risk. In other words, in companies where the risk of financial information disclosure is higher, the quality of corporate governance has been lower. This result implies that a strong corporate governance system can increase disclosure quality. Companies with a higher level of financial information disclosure have a lower risk of information disclosure, which can arise from an effective corporate governance system. Generally, improving corporate governance in a company causes information asymmetry and reduces agency costs and information search costs and increases information transparency because good corporate governance makes managers of a company, in addition to the motivation of earning their own profit, try to advance the interests of investors and the value of the company (Cheng et al. 2019). If the corporate governance system is stronger, the financial information disclosure is higher,

and the information transparency is more. In addition, during the COVID-19 pandemic, companies can improve the quality of their financial reporting by improving corporate governance (mainly by increasing the board of directors) (Hsu and Yang 2022). The quality of financial information disclosure increases with the increase incorporate governance. Therefore, the second hypothesis is confirmed in the negative direction.

We used several sensitivity tests to confirm the robustness of our findings, and we obtained robust results that indicate that companies in the growth and maturity stages have the highest investment efficiency. This will increase with the improvement of corporate governance in the mentioned companies (Ahmed et al. 2020). In addition, the life cycle affects the quality of financial information disclosure and corporate governance (Habib and Hasan 2019). This confirms that improvement in corporate governance leads to an increase in investment efficiency and a reduction in the risk of financial information disclosure in companies in growth and maturity stages. Furthermore, to test the hypotheses, we surveyed the size of the companies. Research shows that investment efficiency increases with the firm size (Luo et al. 2022), and investment efficiency in large companies is more than in small companies. Good corporate governance increases investment efficiency in these companies. In addition, firm size affects the risk of financial information disclosure (Apriliani 2018). In larger companies, good corporate governance reduces the financial information disclosure risk, while improving corporate governance in smaller companies increases the risk of financial information disclosure and reduces investment efficiency.

Our results support the view that improving corporate governance increases investment efficiency and reduces the financial information disclosure risk, which applies to large firms and firms in the growth and maturity stages. Currently, under the unfavorable situations of Iran's economy, managers and board members should institutionalize corporate governance mechanisms in their companies to increase investment efficiency and reduce the financial information disclosure risk. Such an action reduces the potential threats of bankruptcy and financial crisis. The influential effects of corporate governance are highlighted in the decision making of investors and capital market activists to invest and to prevent the distortion of financial information. It is suggested that shareholders and analysts pay attention to this important matter before investing in companies. Moreover, this study points out that investors and shareholders should pay attention to the firm size and the life cycle of the firms in order to estimate their investment risk because the investment risk and financial information disclosure risk are high in small firms and firms that are in their decline stages due to the possibility of manipulation in the financial statements and the investment inefficiency.

Based on the results of the study and considering the sanctions imposed in Iran, it is suggested that future studies analyze and review the influence of corporate governance on the two variables of investment efficiency and financial information disclosure risk, by considering before and after the period of imposition of sanctions. Moreover, from the limitations of this article, if there is a change in the method of measuring the variables, different results may be obtained from the current findings of the study. In addition, this study was carried out in companies admitted to the Tehran Stock Exchange. Therefore, generalizing it to other communities (such as OTC companies or investment companies) should be performed with full caution, and with the change in the time period, the results of the research may change.

**Author Contributions:** Conceptualization, S.M.K.; methodology, and M.M.S.; software, and M.M.S.; vali and M.M.S. dation, S.M.K.; formal analysis, S.M.K. All authors have read and agreed to the published version of the manuscript.

**Funding:** This research received no external funding.

**Data Availability Statement:** The data will be available at request.

**Conflicts of Interest:** The authors declare no conflict of interest.

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
