# Peer review of "The Role of Corporate Governance in Investment Efficiency and Financial Information Disclosure Risk in Companies Listed on the Tehran Stock Exchange"

_jrfm, doi:10.3390/jrfm15120577_

Round 1

Reviewer 1 Report

Article:

Investigating the Role of Corporate Governance in Investment Efficiency and Financial Disclosure Risk in Companies Listed on the Tehran Stock Exchange

After reviewing, I have some comment as follow:

-          The author should revise the abstract. The abstract should introduce the research objective, methodology and main finding. The motivation should not be in the abstract. There, the author should delete the paragraph “With the recent financial scandals around…less risk of financial information disclosure” or move it to the introduction section.

-          In the introduction, this section has not shown research gaps, the authors need to present an overview of previous studies to see research gaps as well as clearly state the contributions of the research.

-          In the literature review, the authors have not presented supporting theories for developing research hypotheses. In addition, this section lacks updating, especially research on corporate governance in developing and emerging countries. Therefore, I suggest the authors update some of the studies as follows: Basiruddin and Ahmed (2019); Dang and Nguyen (2021); Maulidi et al. (2022); Ballester et al. (2020); Nguyen (2021, 2022); Raouf and Ahmed (2020)

-          Section 3.1 should introduce the data source. In this section, the author must explain why selected the period 2015-2020, why not earlier to make larger data?

-          Section 3.2, the author must explain why you have these models? why do you have the dependent variables, on which theory or previous studies, and why? You can't give a model without any explanation. Also, variables need to be introduced explicitly, not to say "method of measuring the variables is presented in the appendix section". Also, the variables in the appendix are not exhaustive, e.g. what is INVEFF and FIDR? It is extremely important to clearly state the measurement of these variables. The authors must clarify in this part, should not be in the appendix. In addition, the authors must explain the estimation methods for these models.

-          Relating corporate governance, there is a potential endogeneity problem (Wintoki et al., 2012). I suggest the authors applied GMM method as another robustness test.

-          In conclusion, the authors need to present research limitations as well as suggestions for future research

References

-                      Ballester, L., González-Urteaga, A., & Martínez, B. (2020). The role of internal corporate governance mechanisms on default risk: A systematic review for different institutional settings. Research in International Business and Finance, 54, 101293.

-                      Basiruddin, R., & Ahmed, H. (2019). Corporate governance and Shariah non-compliant risk in Islamic banks: evidence from Southeast Asia. Corporate Governance: The International Journal of Business in Society.

-                      Dang, V. C., & Nguyen, Q. K. (2021). Internal corporate governance and stock price crash risk: evidence from Vietnam. Journal of Sustainable Finance & Investment, 1-18. doi:10.1080/20430795.2021.2006128

-                      Maulidi, A., Shonhadji, N., Sari, R. P., Nuswantara, D. A., & Widuri, R. (2022). Are female CFOs more ethical to the occurrences of financial reporting fraud? Theoretical and empirical evidence from cross-listed firms in the US. Journal of Financial Crime(ahead-of-print).

-                      Nguyen, Q. K. (2021). Oversight of bank risk-taking by audit committees and Sharia committees: conventional vs Islamic banks. Heliyon, 7(8), e07798.

-                      Nguyen, Q. K. (2022). Audit committee structure, institutional quality, and bank stability: evidence from ASEAN countries. Finance Research Letters, 46, 102369.

-                      Raouf, H., & Ahmed, H. (2020). Risk governance and financial stability: A comparative study of conventional and Islamic banks in the GCC. Global Finance Journal, 100599.

-                      Wintoki, M. B., Linck, J. S., & Netter, J. M. (2012). Endogeneity and the dynamics of internal corporate governance. Journal of financial economics, 105(3), 581-606.

Author Response

Response to Reviewer 1 Comments

Point  1. The author should revise the abstract. The abstract should introduce the research objective, methodology and main finding. The motivation should not be in the abstract. There. The author should delete the paragraph “With the recent financial scandals around… less risk of financial information disclosure” or move it to the introduction section.

 Response1 :

It is modified (page 1).

Point  2. In the introduction, this section has not shown research gaps, the authors need to present an overview of previous studies to see research gaps as well as clearly state the contributions of the research.

 Response2:

.It is added (page 4).

Point  3. In the literature review, the authors have not presented supporting theories for developing research hypotheses. In addition, this section lacks updating, especially research on corporate governance in developing and emerging countries. Therefore, I suggest the authors update some of the studies as follows: Basiruddin and Ahmed (2019); Ballester et al. (2020), Nguyen (2021,2022);

Response 3: Some researches about the corporate governance are added (page 4).

Point 4 . Section 3.1 should introduce the data source. In this section, the author must explain why selected the period 2015-2020, why not earlier to make larger data?

Response 4: It is explained in the article (page 7).

 Point 5. Section 3.2, the author must explain why you have these models? Why do you have the dependent variables, on which theory or previous studies, and why? You can’t give a model without any explanation. Also, variables need to be introduced explicitly, not to say “method of measuring the variables is presented in the appendix section”. Also, the variables in the appendix are not exhaustive, e.g. what is INEFF and FIDR? It is extremely important to clearly state the measurement of these variables. The authors must clarify in this part, should not be in the appendix. In addition, the authors must explain the estimation methods for these models.

Response 5 : It is modified (page 7 to 11).

Point 6 . Relating corporate governance, there is a potential endogeneity problem (Wintoki et al., 2012). I suggest the author applied GMM method as another robustness test.

Response6 : Our article is about the impact of corporate governance on investment efficiency and the financial information disclosure risk. However, GMM method in the suggested article is more about the survey of corporate governance mechanisms, which we think that it is not related to our article.

 Point 7 :In conclusion, the authors need to present research limitations as well as suggestions for future research.

Response 7 : It is added (page 21 and 22).

Reviewer 2 Report

Dear authors, I consider your work valuable and recommend it for publication after minor English proofing (for example, page 2, line 69). 

Congratulations!

Author Response

No comment

Reviewer 3 Report

1.      Abstract should be shorter and more concise.

2.      The paper requires thorough proofreading. Language should be improved.

3.      The Authors should stress key novelties as compared to existing literature. The topic has been widely discussed in the literature. Where is the value added?

4.      The Authors should present their justification for the methodology chosen. 

5.      Is the panel in fact balanced? The authors state “The results of the survey of 140 21 companies listed on the Tehran Stock Exchange (including 980 year-company observations) 22 from2015 to 2021 indicate that investment efficiency has increased by increasing the quality of 23 corporate governance.”

6.      Fundamental terms used in the research (Investment Efficiency, Financial Information Disclosure Risk, Corporate Governance) should be clearly defined in the text, including Introduction.

Author Response

Response to Reviewer 3 Comments

Point 1 : Abstract: The abstract is ok. In my opinion this sentence should be delete – “(including 980 year-company observations).

Response 1: It is modified (page 1).

Pointe 2: Introduction: the introduction provides sufficient background to the work. In my opinion authors should write whether the Covid pandemic had an impact on the research. The period under study is 2020-2021, which is the period of a pandemic that had an impact many issues. It is worth introducing new literature items to better show the impact of the pandemic on the analyzed phenomenon.

Below is a proposal of articles: Sadowski, A., Galar, Z., Walasek, R. et al. Big data insight on mobility during the Covid-19 pandemic lockdown. J Big Data 8, 78 (2021).

Salehi, M., Ammar Ajel, R. and Zimon, G (2022), “The relationship between corporate governance and financial reporting transparency”, Journal of Financial Reporting and Accounting.

Response 2 : The suggested articles were not related to our article. Therefore, we add another articles about the impact of cooperate governance during Covid-19 pandemic (page 6).

Pointe 3: Literature Review: The literature review should be literature should be improved.

Authors should supplement the literature with the latest literature.

Response 3: It is modified (page 4).

Point 4: Methodology: this section is well structures.

Response 4: Result: I have no comments on the results.

Point 5: Conclusion and Discussion: Authors should refer to Covid period publications when creating discussions, after all they study the period that covers the years 2020-2021.

Unless Covid-19 had no effect on the results.

Response 5 : We add some researches about the impact of cooperate governance during Covid-19 pandemic in Conclusion (page 20 and 21).

Reviewer 4 Report

Dear Authors,

I am sending comments to the article:
Investigating the Role of Corporate Governance in Investment Efficiency
and Financial Disclosure Risk in Companies Listed on the Tehran Stock Exchange”

Abstract:

The abstract is ok. In my opinion this sentence should be delete – “(including 980 year-company observations)”

Introduction:

The introduction provides sufficient background to the work.
In my opinion authors should write whether the Covid pandemic had an impact on the research.
The period under study is 2020-2021, which is the period of a pandemic that had an impact on many issues.
It is worth introducing new literature items to better show the impact of the pandemic on the analyzed phenomenon.
Below is a proposal of articles:
Sadowski, A., Galar, Z., Walasek, R. et al. Big data insight on global mobility
during the Covid-19 pandemic lockdown.
J Big Data 8, 78 (2021). https://doi.org/10.1186/s40537-021-00474-2

Salehi, M., Ammar Ajel, R. and Zimon, G. (2022), "The relationship between corporate governance and financial reporting transparency", Journal of Financial Reporting and Accounting, Vol. ahead-of-print No. ahead-of-print. https://doi.org/10.1108/JFRA-04-2021-0102

Literature Review:

The literature review should be literature should be improved.
Authors should supplement the literature with the latest literature

Methodology:

This section is well structured.

Result

I have no comments on the results

Conclusion and Discussion:

Authors should refer to Covid period publications when creating discussions,
after all, they study the period that covers the years 2020-2021.
Unless Covid 19 had no effect on the results.
  Sincerely  

Author Response

Response to Reviewer 4 Comments

Pointe 1 :Abstract should be shorter and more concise.

Response 1 :It is modified (page 1).

Pointe 2: The paper requires thorough proofreading. Language should be improved.

Response 2 :It is modified.

Pointe 3:The Authors should stress key novelties as compared to existing literature. The topic has been widely discussed in the literature. Where is the value added?

Response 3 :It is added (page 4).

Pointe 4: The Authors should present their justification for the methodology chosen.

Response 4 :It is added (page 7).

Pointe5 :Is the panel in fact balanced? The authors state “The results of the survey of 140 21 companies listed on the Tehran Stock Exchange (including 980 year-company observations) 22 from2015 to 2021 indicate that investment efficiency has increased by increasing the quality of 23 corporate governance.”

Response 5 :It is modified.

Pointe6: . Fundamental terms used in the research (Investment Efficiency, Financial Information Disclosure Risk, Corporate Governance) should be clearly defined in the text, including Introduction.

Response 6 :The fundamental terms are defined clearly and completely in “Literature Review” (page 4, 5 and 6).

Round 2

Reviewer 1 Report

The revision is satisfied. 

Author Response

 English language and style are fine/minor spell check required. English language and style are fine/minor spell check required.

Response 1 : It is modified and done.

Reviewer 3 Report

1.      The paper requires thorough proofreading. Language should be improved. In many places spaces are missing and words are combined

2.      Hypotheses could be more precisely defined.

Eg. Hypothesis 1: There is a (statistically? positive?) significant relationship between corporate governance and investment efficiency.

3.      Is the panel in fact balanced? The authors state “The results of the survey of 140 21 companies listed on the Tehran Stock Exchange (including 980 year-company observations) 22 from2015 to 2021 indicate that investment efficiency has increased by increasing the quality of 23 corporate governance.”

4.      Low R2 in regression models (eg. 0.27 in Table 7) should be addressed.

5.      Presentation of tables with regression results should be standardized eg. no need to provide detailed reg. function in first line

6.      Fundamental terms used in the research (Investment Efficiency, Financial Information Disclosure Risk, Corporate Governance) should be clearly defined in the text, including Introduction.

Author Response

  1. The paper requires thorough proofreading. Language should be improved. In many places spaces are missing and words are combined.

Response1: It is modified and done.

  1. Hypotheses could be more precisely defined. Eg. Hypothesis 1: there is a (statistically? positive?) significant relationship between corporate governance and investment efficiency.

Response 2:

It is modified (page 6, 7, 13 and 14).

  1. Is the panel in fact balanced? The authors state “The results of the survey of 140 21 companies listed on the Tehran Stock Exchange (including 980 year-company observations) 22 from2015 to 2021 indicate that investment efficiency has increased by increasing the quality of 23 corporate governance.”

 Response 3: We did not mention that “21 companies listed on the Tehran Stock Exchange” or “the 22 from2015 to 2021 indicate that investment efficiency has increased by increasing the quality of  23 corporate governance”.

  1. Low R2 in regression models (eg. 0.27 in table 7) should be addressed.

Response 4: It is added (page 13).

  1. Presentation of tables with regression results should be standardized eg. No need to provide detailed reg. function in first line.

Response 5: It is modified (page 11, 12, 14, 15, 16, 17, 18and 19).

  1. Fundamental terms used in the research (Investment Efficiency, Financial Information Disclosure Risk, Corporate Governance) should be clearly defined in the text, including Introduction.

Response 6: It is added (page 1 and 2).
